# Dream to Control: Learning Behaviors by Latent Imagination

**Danijar Hafner** [*]
University of Toronto
Google Brain

**Timothy Lillicrap**
DeepMind

**Jimmy Ba**
University of Toronto

**Mohammad Norouzi**
Google Brain

### Abstract

Learned world models summarize an agent's experience to facilitate learning complex behaviors. While learning world models from high-dimensional sensory inputs is becoming feasible through deep learning, there are many potential ways for deriving behaviors from them. We present Dreamer, a reinforcement learning agent that solves long-horizon tasks from images purely by latent imagination. We efficiently learn behaviors by propagating analytic gradients of learned state values back through trajectories imagined in the compact state space of a learned world model. On 20 challenging visual control tasks, Dreamer exceeds existing approaches in data-efficiency, computation time, and final performance.

## 1 Introduction

Intelligent agents can achieve goals in complex environments even though they never encounter the exact same situation twice. This ability requires building representations of the world from past experience that enable generalization to novel situations. World models offer an explicit way to represent an agent's knowledge about the world in a parametric model that can make predictions about the future.

When the sensory inputs are high-dimensional images, latent dynamics models can abstract observations to predict forward in compact state spaces (Watter et al., 2015; Oh et al., 2017; Gregor et al., 2019). Compared to predictions in image space, latent states have a small memory footprint that enables imagining thousands of trajectories in parallel. Learning effective latent dynamics models is becoming feasible through advances in deep learning and latent variable models (Krishnan et al., 2015; Karl et al., 2016; Doerr et al., 2018; Buesing et al., 2018).

Behaviors can be derived from dynamics models in many ways. Often, imagined rewards are maximized with a parametric policy (Sutton, 1991; Ha and Schmidhuber, 2018; Zhang et al., 2019) or by online planning (Chua et al., 2018; Hafner et al., 2018). However, considering only rewards within a fixed imagination horizon results in shortsighted behaviors (Wang et al., 2019). Moreover, prior work commonly resorts to derivative-free optimization for robustness to model errors (Ebert et al., 2017; Chua et al., 2018; Parmas et al., 2019), rather than leveraging analytic gradients offered by neural network dynamics (Henaff et al., 2019; Srinivas et al., 2018).

We present Dreamer, an agent that learns long-horizon behaviors from images purely by latent imagination. A novel actor critic algorithm accounts for rewards beyond the imagination horizon while making efficient use of the neural network dynamics. For this, we predict state values and actions in the learned latent space as summarized in Figure 1. The values optimize Bellman consistency for imagined rewards and the policy maximizes the values by propagating their analytic gradients back through the dynamics.

In comparison to actor critic algorithms that learn online or by experience replay (Lillicrap et al., 2015; Mnih et al., 2016; Schulman et al., 2017; Haarnoja et al., 2018; Lee et al., 2019), world models can interpolate past experience and offer analytic gradients of multi-step returns for efficient policy optimization.

Dataset of Experience

Learned Latent Dynamics

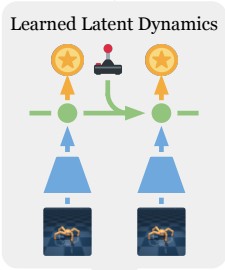

Value and Action Learned by Latent Imagination

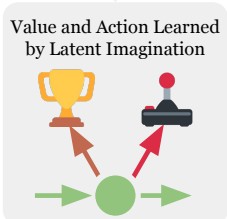

Figure 1: Dreamer learns a world model from past experience and efficiently learns farsighted behaviors in its latent space by backpropagating value estimates back through imagined trajectories.

---

[*]Correspondence to: Danijar Hafner <mail@danijar.com>.

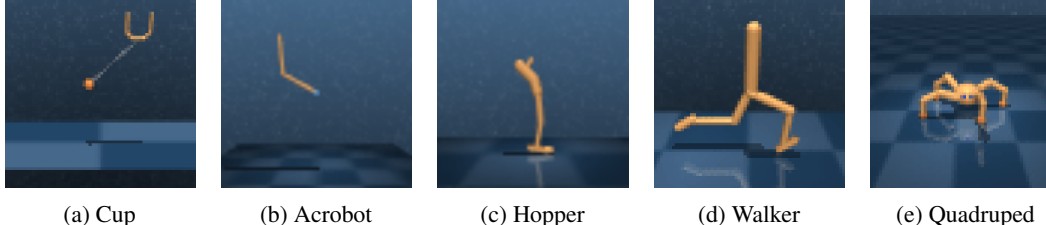

| (a) Cup | (b) Acrobot | (c) Hopper | (d) Walker | (e) Quadruped |

Figure 2: Image observations for 5 of the 20 visual control tasks used in our experiments. The tasks pose a variety of challenges including contact dynamics, sparse rewards, many degrees of freedom, and 3D environments. Several of these tasks could previously not be solved through world models.

The key contributions of this paper are summarized as follows:

- **Learning long-horizon behaviors by latent imagination**  Model-based agents can be short-sighted if they use a finite imagination horizon. We approach this limitation by predicting both actions and state values. Training purely by imagination in a latent space lets us efficiently learn the policy by propagating analytic value gradients back through the latent dynamics.

- **Empirical performance for visual control**  We pair Dreamer with existing representation learning methods and evaluate it on the DeepMind Control Suite with image inputs, illustrated in Figure 2. Using the same hyper parameters for all tasks, Dreamer exceeds previous model-based and model-free agents in terms of data-efficiency, computation time, and final performance.

## 2 CONTROL WITH WORLD MODELS

**Reinforcement learning**  We formulate visual control as a partially observable Markov decision process (POMDP) with discrete time step $t \in [1; T]$, continuous vector-valued actions $a_t \sim p(a_t \mid o_{\leq t}, a_{<t})$ generated by the agent, and high-dimensional observations and scalar rewards $o_t, r_t \sim p(o_t, r_t \mid o_{<t}, a_{<t})$ generated by the unknown environment. The goal is to develop an agent that maximizes the expected sum of rewards $\mathrm{E}_p\left(\sum_{t=1}^T r_t\right)$. Figure 2 shows a selection of our tasks.

**Agent components**  The classical components of agents that learn in imagination are dynamics learning, behavior learning, and environment interaction (Sutton, 1991). In the case of Dreamer, the behavior is learned by predicting hypothetical trajectories in the compact latent space of the world model. As outlined in Figure 3 and detailed in Algorithm 1, Dreamer performs the following operations throughout the agent's life time, either interleaved or in parallel:

- Learning the latent dynamics model from the dataset of past experience to predict future rewards from actions and past observations. Any learning objective for the world model can be incorporated with Dreamer. We review existing methods for learning latent dynamics in Section 4.

- Learning action and value models from predicted latent trajectories, as described in Section 3. The value model optimizes Bellman consistency for imagined rewards and the action model is updated by propagating gradients of value estimates back through the neural network dynamics.

- Executing the learned action model in the world to collect new experience for growing the dataset.

**Latent dynamics**  Dreamer uses a latent dynamics model that consists of three components. The representation model encodes observations and actions to create continuous vector-valued model states $s_t$ with Markovian transitions (Watter et al., 2015; Zhang et al., 2019; Hafner et al., 2018). The transition model predicts future model states without seeing the corresponding observations that will later cause them. The reward model predicts the rewards given the model states,

$$
\begin{aligned}
\text{Representation model:} \quad & p(s_t \mid s_{t-1}, a_{t-1}, o_t) \\
\text{Transition model:} \quad & q(s_t \mid s_{t-1}, a_{t-1}) \\
\text{Reward model:} \quad & q(r_t \mid s_t).
\end{aligned}
\tag{1}
$$

We use $p$ for distributions that generate samples in the real environment and $q$ for their approximations that enable latent imagination. Specifically, the transition model lets us predict ahead in the compact latent space without having to observe or imagine the corresponding images. This results in a low memory footprint and fast predictions of thousands of imagined trajectories in parallel.

The model mimics a non-linear Kalman filter (Kalman, 1960), latent state space model, or HMM with real-valued states. However, it is conditioned on actions and predicts rewards, allowing the agent to imagine the outcomes of potential action sequences without executing them in the environment.

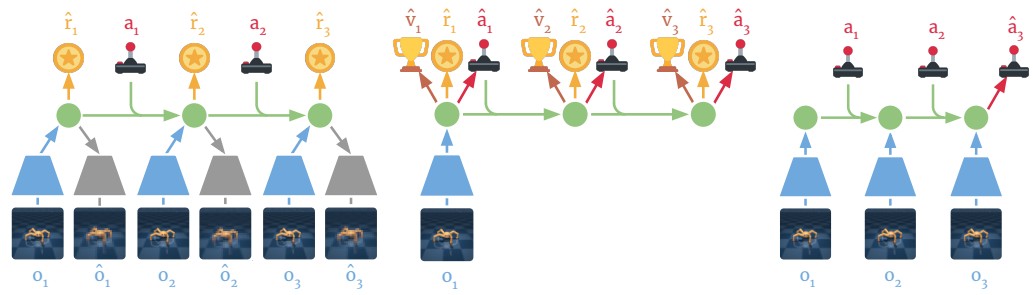

(a) Learn dynamics from experience    (b) Learn behavior in imagination    (c) Act in the environment

Figure 3: Components of Dreamer. (a) From the dataset of past experience, the agent learns to encode observations and actions into compact latent states (●), for example via reconstruction, and predicts environment rewards (⊛). (b) In the compact latent space, Dreamer predicts state values (🏆) and actions (⛏) that maximize future value predictions by propagating gradients back through imagined trajectories. (c) The agent encodes the history of the episode to compute the current model state and predict the next action to execute in the environment. See Algorithm 1 for pseudo code of the agent.

## 3  LEARNING BEHAVIORS BY LATENT IMAGINATION

Dreamer learns long-horizon behaviors in the compact latent space of a learned world model by efficiently leveraging the neural network latent dynamics. For this, we propagate stochastic gradients of multi-step returns through neural network predictions of actions, states, rewards, and values using reparameterization. This section describes the main contribution of our paper.

**Imagination environment**    The latent dynamics define a Markov decision process (MDP; Sutton, 1991) that is fully observed because the compact model states $s_t$ are Markovian. We denote imagined quantities with $\tau$ as the time index. Imagined trajectories start at the true model states $s_t$ of observation sequences drawn from the agent's past experience. They follow predictions of the transition model $s_\tau \sim q(s_\tau \mid s_{\tau-1}, a_{\tau-1})$, reward model $r_\tau \sim q(r_\tau \mid s_\tau)$, and a policy $a_\tau \sim q(a_\tau \mid s_\tau)$. The objective is to maximize expected imagined rewards $\mathrm{E}_q\left(\sum_{\tau=t}^{\infty} \gamma^{\tau-t} r_\tau\right)$ with respect to the policy.

---

**Algorithm 1:** Dreamer

---

Initialize dataset $\mathcal{D}$ with $S$ random seed episodes.
Initialize neural network parameters $\theta, \phi, \psi$ randomly.
**while** *not converged* **do**
   **for** *update step* $c = 1..C$ **do**

      // Dynamics learning
      Draw $B$ data sequences $\{(a_t, o_t, r_t)\}_{t=k}^{k+L} \sim \mathcal{D}$.
      Compute model states $s_t \sim p_\theta(s_t \mid s_{t-1}, a_{t-1}, o_t)$.
      Update $\theta$ using representation learning.

      // Behavior learning
      Imagine trajectories $\{(s_\tau, a_\tau)\}_{\tau=t}^{t+H}$ from each $s_t$.
      Predict rewards $\mathrm{E}\left(q_\theta(r_\tau \mid s_\tau)\right)$ and values $v_\psi(s_\tau)$.
      Compute value estimates $\mathrm{V}_\lambda(s_\tau)$ via Equation 6.
      Update $\phi \leftarrow \phi + \alpha \nabla_\phi \sum_{\tau=t}^{t+H} \mathrm{V}_\lambda(s_\tau)$.
      Update $\psi \leftarrow \psi - \alpha \nabla_\psi \sum_{\tau=t}^{t+H} \frac{1}{2}\left\|v_\psi(s_\tau) - \mathrm{V}_\lambda(s_\tau)\right\|^2$.

   // Environment interaction
   $o_1 \leftarrow$ env.reset()
   **for** *time step* $t = 1..T$ **do**
      Compute $s_t \sim p_\theta(s_t \mid s_{t-1}, a_{t-1}, o_t)$ from history.
      Compute $a_t \sim q_\phi(a_t \mid s_t)$ with the action model.
      Add exploration noise to action.
      $r_t, o_{t+1} \leftarrow$ env.step($a_t$).
   Add experience to dataset $\mathcal{D} \leftarrow \mathcal{D} \cup \{(o_t, a_t, r_t)_{t=1}^{T}\}$.

**Model components**

| | |
|---|---|
| Representation | $p_\theta(s_t \mid s_{t\text{-}1}, a_{t\text{-}1}, o_t)$ |
| Transition | $q_\theta(s_t \mid s_{t\text{-}1}, a_{t\text{-}1})$ |
| Reward | $q_\theta(r_t \mid s_t)$ |
| Action | $q_\phi(a_t \mid s_t)$ |
| Value | $v_\psi(s_t)$ |

**Hyper parameters**

| | |
|---|---|
| Seed episodes | $S$ |
| Collect interval | $C$ |
| Batch size | $B$ |
| Sequence length | $L$ |
| Imagination horizon | $H$ |
| Learning rate | $\alpha$ |

---

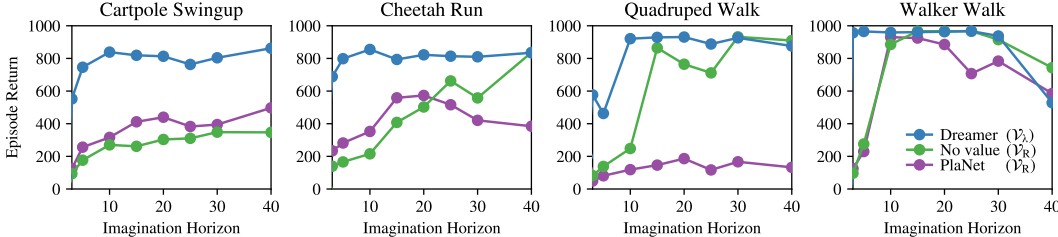

Figure 4: Imagination horizons. We compare the final performance of Dreamer, learning an action model without value prediction, and online planning using PlaNet. Learning a state value model to estimate rewards beyond the imagination horizon makes Dreamer more robust to the horizon length. The agents use pixel reconstruction for representation learning and an action repeat of $R = 2$.

**Action and value models**  Consider imagined trajectories with a finite horizon $H$. Dreamer uses an actor critic approach to learn behaviors that consider rewards beyond the horizon. We learn an action model and a value model in the latent space of the world model for this. The action model implements the policy and aims to predict actions that solve the imagination environment. The value model estimates the expected imagined rewards that the action model achieves from each state $s_\tau$,

$$
\begin{aligned}
\text{Action model:} \quad & a_\tau \sim q_\phi(a_\tau \mid s_\tau) \\
\text{Value model:} \quad & v_\psi(s_\tau) \approx \mathrm{E}_{q(\cdot \mid s_\tau)}\left(\sum_{\tau=t}^{t+H} \gamma^{\tau-t} r_\tau\right).
\end{aligned}
\tag{2}
$$

The action and value models are trained cooperatively as typical in policy iteration: the action model aims to maximize an estimate of the value, while the value model aims to match an estimate of the value that changes as the action model changes.

We use dense neural networks for the action and value models with parameters $\phi$ and $\psi$, respectively. The action model outputs a tanh-transformed Gaussian (Haarnoja et al., 2018) with sufficient statistics predicted by the neural network. This allows for reparameterized sampling (Kingma and Welling, 2013; Rezende et al., 2014) that views sampled actions as deterministically dependent on the neural network output, allowing us to backpropagate analytic gradients through the sampling operation,

$$
a_\tau = \tanh\left(\mu_\phi(s_\tau) + \sigma_\phi(s_\tau)\, \epsilon\right), \quad \epsilon \sim \text{Normal}(0, \mathbb{I}).
\tag{3}
$$

**Value estimation**  To learn the action and value models, we need to estimate the state values of imagined trajectories $\{s_\tau, a_\tau, r_\tau\}_{\tau=t}^{t+H}$. These trajectories branch off of the model states $s_t$ of sequence batches drawn from the agent's dataset of experience and predict forward for the imagination horizon $H$ using actions sampled from the action model. State values can be estimated in multiple ways that trade off bias and variance (Sutton and Barto, 2018),

$$
\mathrm{V}_{\mathrm{R}}(s_\tau) \doteq \mathrm{E}_{q_\theta, q_\phi}\left(\sum_{n=\tau}^{t+H} r_n\right),
\tag{4}
$$

$$
\mathrm{V}_{\mathrm{N}}^k(s_\tau) \doteq \mathrm{E}_{q_\theta, q_\phi}\left(\sum_{n=\tau}^{h-1} \gamma^{n-\tau} r_n + \gamma^{h-\tau} v_\psi(s_h)\right) \quad \text{with} \quad h = \min(\tau + k, t + H),
\tag{5}
$$

$$
\mathrm{V}_\lambda(s_\tau) \doteq (1 - \lambda) \sum_{n=1}^{H-1} \lambda^{n-1} \mathrm{V}_{\mathrm{N}}^n(s_\tau) + \lambda^{H-1} \mathrm{V}_{\mathrm{N}}^H(s_\tau),
\tag{6}
$$

where the expectations are estimated under the imagined trajectories. $\mathrm{V}_{\mathrm{R}}$ simply sums the rewards from $\tau$ until the horizon and ignores rewards beyond it. This allows learning the action model without a value model, an ablation we compare to in our experiments. $\mathrm{V}_{\mathrm{N}}^k$ estimates rewards beyond $k$ steps with the learned value model. Dreamer uses $\mathrm{V}_\lambda$, an exponentially-weighted average of the estimates for different $k$ to balance bias and variance. Figure 4 shows that learning a value model in imagination enables Dreamer to solve long-horizon tasks while being robust to the imagination horizon. The experimental details and results on all tasks are described in Section 6.

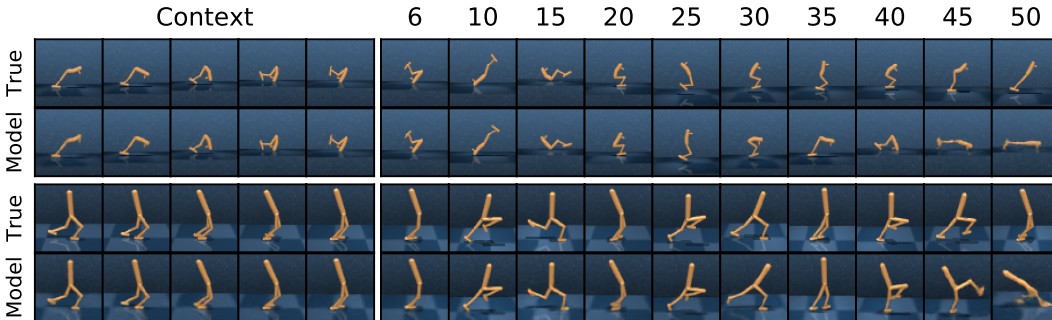

Figure 5: Reconstructions of long-term predictions. We apply the representation model to the first 5 images of two hold-out trajectories and predict forward for 45 steps using the latent dynamics, given only the actions. The recurrent state space model (RSSM; Hafner et al., 2018) performs accurate long-term predictions, enabling Dreamer to learn successful behaviors in a compact latent space.

**Learning objective** To update the action and value models, we first compute the value estimates $V_\lambda(s_\tau)$ for all states $s_\tau$ along the imagined trajectories. The objective for the action model $q_\phi(a_\tau \mid s_\tau)$ is to predict actions that result in state trajectories with high value estimates. The objective for the value model $v_\psi(s_\tau)$, in turn, is to regress the value estimates,

$$\max_\phi E_{q_\theta, q_\phi}\left(\sum_{\tau=t}^{t+H} V_\lambda(s_\tau)\right), \qquad (7) \qquad \min_\psi E_{q_\theta, q_\phi}\left(\sum_{\tau=t}^{t+H} \frac{1}{2}\Big\|v_\psi(s_\tau) - V_\lambda(s_\tau))\Big\|^2\right). \qquad (8)$$

The value model is updated to regress the targets, around which we stop the gradient as typical (Sutton and Barto, 2018). The action model uses analytic gradients through the learned dynamics to maximize the value estimates. To understand this, we note that the value estimates depend on the reward and value predictions, which depend on the imagined states, which in turn depend on the imagined actions. Since all steps are implemented as neural networks, we analytically compute $\nabla_\phi E_{q_\theta, q_\phi}\left(\sum_{\tau=t}^{t+H} V_\lambda(s_\tau)\right)$ by stochastic backpropagation (Kingma and Welling, 2013; Rezende et al., 2014). We use reparameterization for continuous actions and latent states and straight-through gradients (Bengio et al., 2013) for discrete actions. The world model is fixed while learning behaviors.

In tasks with early termination, the world model also predicts the discount factor from each latent state to weigh the time steps in Equations 7 and 8 by the cumulative product of the predicted discount factors, so terms are weighted down based on how likely the imagined trajectory would have ended.

**Comparison to actor critic methods** Agents using Reinforce gradients (Williams, 1992), such as A3C and PPO (Mnih et al., 2016; Schulman et al., 2017), employ value baselines to reduce gradient variance, while Dreamer backpropagates through the value model. This is similar to deterministic or reparameterized actor critics (Silver et al., 2014), such as DDPG and SAC (Lillicrap et al., 2015; Haarnoja et al., 2018). However, these do not leverage gradients through transitions and only maximize immediate Q-values. MVE and STEVE (Feinberg et al., 2018; Buckman et al., 2018) extend them to multi-step Q-learning with learned dynamics to provide more accurate Q-value targets. We predict state values, which is sufficient for policy optimization since we backpropagate through the dynamics. Refer to Section 5 for a more detailed comparison to related work.

## 4 LEARNING LATENT DYNAMICS

Learning behaviors in imagination requires a world model that generalizes well. We focus on latent dynamics models that predict forward in a compact latent space, facilitating long-term predictions and allowing the agent to imagine thousands of trajectories in parallel. Several objectives for learning representations for control have been proposed (Watter et al., 2015; Jaderberg et al., 2016; Oord et al., 2018; Eslami et al., 2018). We review three approaches for learning representations to use with Dreamer: reward prediction, image reconstruction, and contrastive estimation.

**Reward prediction** Latent imagination requires a representation model $p(s_t \mid s_{t-1}, a_{t-1}, o_t)$, transition model $q(s_t \mid s_{t-1}, a_{t-1},)$, and reward model $q(r_t \mid s_t)$, as described in Section 2. In principle, this could be achieved by simply learning to predict future rewards given actions and past observations (Oh et al., 2017; Gelada et al., 2019; Schrittwieser et al., 2019). With a large and diverse dataset, such representations should be sufficient for solving a control task. However, with a finite dataset and especially when rewards are sparse, learning about observations that correlate with rewards is likely to improve the world model (Jaderberg et al., 2016; Gregor et al., 2019).

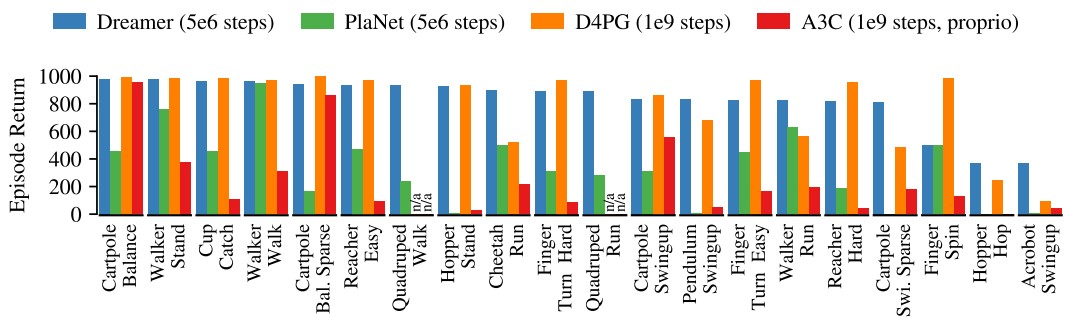

Figure 6: Performance comparison to existing methods. Dreamer inherits the data-efficiency of PlaNet while exceeding the asymptotic performance of the best model-free agents. After $5 \times 10^6$ environment steps, Dreamer reaches an average performance of 823 across tasks, compared to PlaNet at 332 and the top model-free D4PG agent at 786 after $10^9$ steps. Results are averages over 5 seeds.

**Reconstruction**  We first describe the world model used by PlaNet (Hafner et al., 2018) that learns latent dynamics by reconstructing images as shown in Figure 3a. The world model consists of the following components, where the observation model is only used to provide a learning signal,

$$
\begin{array}{lll}
\text{Representation model:} & p_\theta(s_t \mid s_{t-1}, a_{t-1}, o_t) \\
\text{Observation model:} & q_\theta(o_t \mid s_t) \\
\text{Reward model:} & q_\theta(r_t \mid s_t) \\
\text{Transition model:} & q_\theta(s_t \mid s_{t-1}, a_{t-1}).
\end{array}
\tag{9}
$$

The components are optimized jointly to increase the variational lower bound (ELBO; Jordan et al., 1999) or more generally the variational information bottleneck (VIB; Tishby et al., 2000; Alemi et al., 2016). As derived in Appendix B, the bound includes reconstruction terms for observations and rewards and a KL regularizer. The expectation is taken under the dataset and representation model,

$$
\mathcal{J}_{\text{REC}} \doteq \mathrm{E}_p\bigg( \sum_t \Big( \mathcal{J}_O^t + \mathcal{J}_R^t + \mathcal{J}_D^t \Big) \bigg) + \text{const} \qquad \mathcal{J}_O^t \doteq \ln q(o_t \mid s_t)
$$
$$
\mathcal{J}_R^t \doteq \ln q(r_t \mid s_t) \qquad \mathcal{J}_D^t \doteq -\beta \, \mathrm{KL}\big( p(s_t \mid s_{t-1}, a_{t-1}, o_t) \, \big\| \, q(s_t \mid s_{t-1}, a_{t-1}) \big).
\tag{10}
$$

We implement the transition model as a recurrent state space model (RSSM; Hafner et al., 2018), the representation model by combining the RSSM with a convolutional neural network (CNN; LeCun et al., 1989) applied to the image observation, the observation model as a transposed CNN, and the reward model as a dense network. The combined parameter vector $\theta$ is updated by stochastic backpropagation (Kingma and Welling, 2013; Rezende et al., 2014). Figure 5 shows video predictions of this model. We refer to Appendix A and Hafner et al. (2018) model details.

**Contrastive estimation**  Predicting pixels can require high model capacity. We can also encourage mutual information between model states and observations by instead predicting the states from the images (Guo et al., 2018). This replaces the observation model with a state model,

$$
\text{State model:} \qquad q_\theta(s_t \mid o_t).
\tag{11}
$$

While the reconstruction objective used the fact that the observation marginal is a constant, we now face the state marginal. As shown in Appendix B, this can be estimated via noise contrastive estimation (NCE; Gutmann and Hyvärinen, 2010; Oord et al., 2018) by averaging the state model over observations $o'$ of the current sequence batch. Intuitively, $q(s_t \mid o_t)$ makes the state predictable from the current image while $\ln \sum_{o'} q(s_t \mid o')$ keeps it diverse to prevent collapse,

$$
\mathcal{J}_{\text{NCE}} \doteq \mathrm{E}\bigg( \sum_t \Big( \mathcal{J}_S^t + \mathcal{J}_R^t + \mathcal{J}_D^t \Big) \bigg) \quad \mathcal{J}_S^t \doteq \ln q(s_t \mid o_t) - \ln \bigg( \sum_{o'} q(s_t \mid o') \bigg).
\tag{12}
$$

We implement the state model as a CNN and again optimize the bound with respect to the combined parameter vector $\theta$ using stochastic backpropagation. While avoiding pixel prediction, the amount of information this bound can extract efficiently is limited (McAllester and Statos, 2018). We empirically compare reward, reconstruction, and contrastive objectives in our experiments in Figure 8.

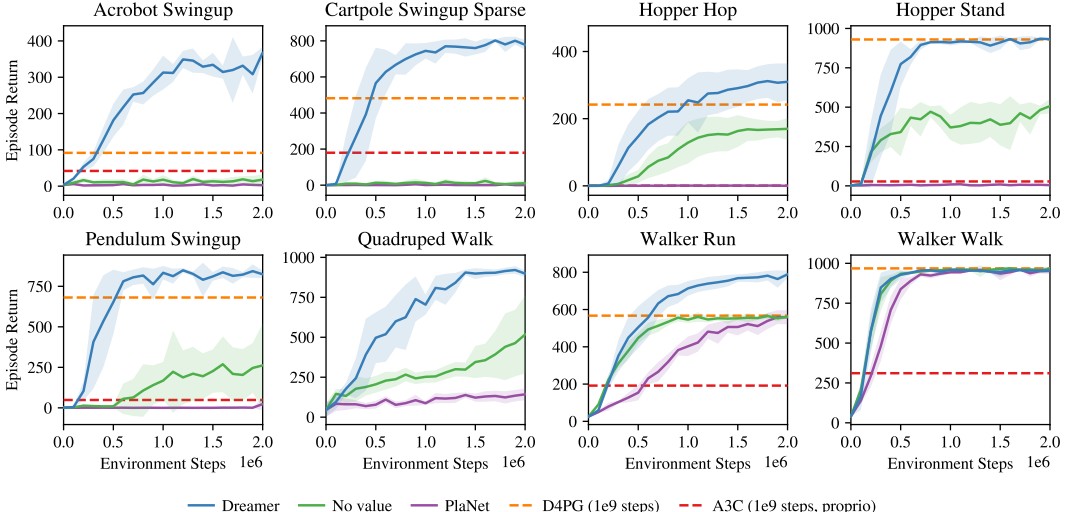

Figure 7: Dreamer succeeds at visual control tasks that require long-horizon credit assignment, such as the acrobot and hopper tasks. Optimizing only imagined rewards within the horizon via an action model or by online planning yields shortsighted behaviors that only succeed in reactive tasks, such as in the walker domain. The performance on all 20 tasks is summarized in Figure 6 and training curves are shown in Appendix D. See Tassa et al. (2018) for performance curves of D4PG and A3C.

## 5 RELATED WORK

Prior works learn latent dynamics for visual control by derivative-free policy learning or online planning, augment model-free agents with multi-step predictions, or use analytic gradients of Q-values or multi-step rewards, often for low-dimensional tasks. In comparison, Dreamer uses analytic gradients to efficiently learn long-horizon behaviors for visual control purely by latent imagination.

**Control with latent dynamics** E2C (Watter et al., 2015) and RCE (Banijamali et al., 2017) embed images to predict forward in a compact space to solve simple tasks. World Models (Ha and Schmidhuber, 2018) learn latent dynamics in a two-stage process to evolve linear controllers in imagination. PlaNet (Hafner et al., 2018) learns them jointly and solves visual locomotion tasks by latent online planning. SOLAR (Zhang et al., 2019) solves robotic tasks via guided policy search in latent space. I2A (Weber et al., 2017) hands imagined trajectories to a model-free policy, while Lee et al. (2019) and Gregor et al. (2019) learn belief representations to accelerate model-free agents.

**Imagined multi-step returns** VPN (Oh et al., 2017), MVE (Feinberg et al., 2018), and STEVE (Buckman et al., 2018) learn dynamics for multi-step Q-learning from a replay buffer. AlphaGo (Silver et al., 2017) combines predictions of actions and state values with planning, assuming access to the true dynamics. Also assuming access to the dynamics, POLO (Lowrey et al., 2018) plans to explore by learning a value ensemble. MuZero (Schrittwieser et al., 2019) learns task-specific reward and value models to solve challenging tasks but requires large amounts of experience. PETS (Chua et al., 2018), VisualMPC (Ebert et al., 2017), and PlaNet (Hafner et al., 2018) plan online using derivative-free optimization. POPLIN (Wang and Ba, 2019) improves over online planning by self-imitation. Piergiovanni et al. (2018) learn robot policies by imagination with a latent dynamics model. Planning with neural network gradients was shown on small problems (Schmidhuber, 1990; Henaff et al., 2018) but has been challenging to scale (Parmas et al., 2019).

**Analytic value gradients** DPG (Silver et al., 2014), DDPG (Lillicrap et al., 2015), and SAC (Haarnoja et al., 2018) leverage gradients of learned immediate action values to learn a policy by experience replay. SVG (Heess et al., 2015) reduces the variance of model-free on-policy algorithms by analytic value gradients of one-step model predictions. Concurrent work by Byravan et al. (2019) uses latent imagination with deterministic models for navigation and manipulation tasks. ME-TRPO (Kurutach et al., 2018) accelerates an otherwise model-free agent via gradients of predicted rewards for proprioceptive inputs. DistGBP (Henaff et al., 2017; 2019) uses model gradients for online planning in simple tasks.

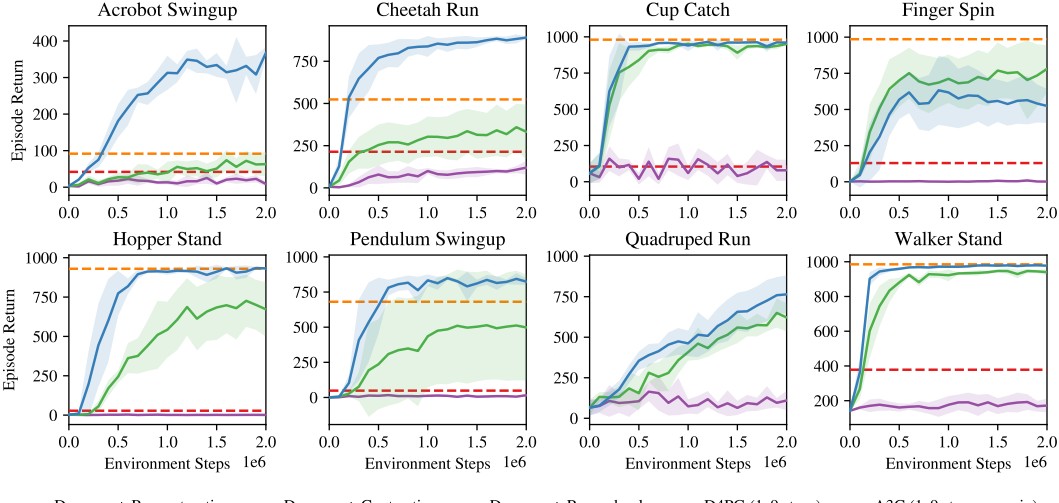

Figure 8: Comparison of representation learning objectives to be used with Dreamer. Pixel reconstruction performs best for the majority of tasks. The contrastive objective solves about half of the tasks, while predicting rewards alone was not sufficient in our experiments. The results suggest that future developments in learning representations are likely to translate into improved task performance for Dreamer. The performance curves for all tasks are included in Appendix E.

## 6 EXPERIMENTS

We experimentally evaluate Dreamer on a variety of control tasks. We designed the experiments to compare Dreamer to current best methods in the literature, and to evaluate its ability to solve tasks with long horizons, continuous actions, discrete actions, and early termination. We further compare the orthogonal choice of learning objective for the world model. The source code for all our experiments and videos of Dreamer are available at https://danijar.com/dreamer.

**Control tasks** We evaluate Dreamer on 20 visual control tasks of the DeepMind Control Suite (Tassa et al., 2018), illustrated in Figure 2. These tasks pose a variety of challenges, including sparse rewards, contact dynamics, and 3D scenes. We selected the tasks on which Tassa et al. (2018) report non-zero performance from image inputs. Agent observations are images of shape $64 \times 64 \times 3$, actions range from 1 to 12 dimensions, rewards range from 0 to 1, episodes last for 1000 steps and have randomized initial states. We use a fixed action repeat of $R = 2$ across tasks. We further evaluate the applicability of Dreamer to discrete actions and early termination on a subset of Atari games (Bellemare et al., 2013) and DeepMind Lab levels (Beattie et al., 2016) as detailed in Appendix C.

**Implementation** Our implementation uses TensorFlow Probability (Dillon et al., 2017). We use a single Nvidia V100 GPU and 10 CPU cores for each training run. The training time for our Dreamer implementation is below 5 hours per $10^6$ environment steps on the control suite, compared to 11 hours for online planning using PlaNet, and the 24 hours used by D4PG to reach similar performance. We use the same hyper parameters across all continuous tasks, and similarly across all discrete tasks, detailed in Appendix A. The world models are learned via reconstruction unless specified.

**Baseline methods** The highest reported performance on the continuous tasks is achieved by D4PG (Barth-Maron et al., 2018), an improved variant of DDPG (Lillicrap et al., 2015) that uses distributed collection, distributional Q-learning, multi-step returns, and prioritized replay. We include the scores for D4PG with pixel inputs and A3C (Mnih et al., 2016) with state inputs from Tassa et al. (2018). PlaNet (Hafner et al., 2018) learns the same world model as Dreamer and selects actions via online planning without an action model and drastically improves over D4PG and A3C in data efficiency. We re-run PlaNet with $R = 2$ for a unified experimental setup. For Atari, we show the final performance of SimPLe (Kaiser et al., 2019), DQN (Mnih et al., 2015) and Rainbow (Hessel et al., 2018) reported by Castro et al. (2018), and for DeepMind Lab that of IMPALA (Espeholt et al., 2018) as a guideline.

**Performance**    To evaluate the performance of Dreamer, we compare it to state-of-the-art reinforcement learning agents. The results are summarized in Figure 6. With an average score of 823 across tasks after $5 \times 10^6$ environment steps, Dreamer exceeds the performance of the strong model-free D4PG agent that achieves an average of 786 within $10^9$ environment steps. At the same time, Dreamer inherits the data-efficiency of PlaNet, confirming that the learned world model can help to generalize from small amounts of experience. The empirical success of Dreamer shows that learning behaviors by latent imagination with world models can outperform top methods based on experience replay.

**Long horizons**    To investigate its ability to learn long-horizon behaviors, we compare Dreamer to alternatives for deriving behaviors from the world model at various horizon lengths. For this, we learn an action model to maximize imagined rewards without a value model and compare to online planning using PlaNet. Figure 4 shows the final performance for different imagination horizons, confirming that the value model makes Dreamer more robust to the horizon and performs well even for short horizons. Performance curves for all 19 tasks with horizon of 20 are shown in Appendix D, where Dreamer outperforms the alternatives on 16 of 20 tasks, with 4 ties.

**Representation learning**    Dreamer can be used with any differentiable dynamics model that predicts future rewards given actions and past observations. Since the representation learning objective is orthogonal to our algorithm, we compare three natural choices described in Section 4: pixel reconstruction, contrastive estimation, and pure reward prediction. Figure 8 shows clear differences in task performance for different representation learning approaches, with pixel reconstruction outperforming contrastive estimation on most tasks. This suggests that future improvements in representation learning are likely to translate to higher task performance with Dreamer. Reward prediction alone was not sufficient in our experiments. Further ablations are included in the appendix of the paper.

## 7    CONCLUSION

We present Dreamer, an agent that learns long-horizon behaviors purely by latent imagination. For this, we propose an actor critic method that optimizes a parametric policy by propagating analytic gradients of multi-step values back through learned latent dynamics. Dreamer outperforms previous methods in data-efficiency, computation time, and final performance on a variety of challenging continuous control tasks with image inputs. We further show that Dreamer is applicable to tasks with discrete actions and early episode termination. Future research on representation learning can likely scale latent imagination to environments of higher visual complexity.

**Acknowledgements**    We thank Simon Kornblith, Benjamin Eysenbach, Ian Fischer, Amy Zhang, Geoffrey Hinton, Shane Gu, Adam Kosiorek, Jacob Buckman, Calvin Luo, and Rishabh Agarwal, and our anonymous reviewers for feedback and discussions. We thank Yuval Tassa for adding the quadruped environment to the control suite.

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

## A    HYPER PARAMETERS

**Model components**    We use the convolutional encoder and decoder networks from Ha and Schmidhuber (2018), the RSSM of Hafner et al. (2018), and implement all other functions as three dense layers of size 300 with ELU activations (Clevert et al., 2015). Distributions in latent space are 30-dimensional diagonal Gaussians. The action model outputs a tanh mean scaled by a factor of 5 and a softplus standard deviation for the Normal distribution that is then transformed using tanh (Haarnoja et al., 2018). The scaling factor allows the agent to saturate the action distribution.

**Learning updates**    We draw batches of 50 sequences of length 50 to train the world model, value model, and action model models using Adam (Kingma and Ba, 2014) with learning rates $6 \times 10^{-4}$, $8 \times 10^{-5}$, $8 \times 10^{-5}$, respectively and scale down gradient norms that exceed 100. We do not scale the KL regularizers ($\beta = 1$) but clip them below 3 free nats as in PlaNet. The imagination horizon is $H = 15$ and the same trajectories are used to update both action and value models. We compute the $V_\lambda$ targets with $\gamma = 0.99$ and $\lambda = 0.95$. We did not find latent overshooting for learning the model, an entropy bonus for the action model, or target networks for the value model necessary.

**Environment interaction**    The dataset is initialized with $S = 5$ episodes collected using random actions. We iterate between 100 training steps and collecting 1 episode by executing the predicted mode action with $\mathrm{Normal}(0, 0.3)$ exploration noise. Instead of manually selecting the action repeat for each environment as in Hafner et al. (2018) and Lee et al. (2019), we fix it to 2 for all environments. See Figure 12 for an assessment of the robustness to different action repeat values.

**Discrete control**    For experiments on Atari games and DeepMind Lab levels, the action model predicts the logits of a categorical distribution. We use straight-through gradients for the sampling step during latent imagination. The action noise is epsilon greedy where $\epsilon$ is linearly scheduled from $0.4 \rightarrow 0.1$ over the first $200,000$ gradient steps. To account for the higher complexity of these tasks, we use an imagination horizon of $H = 10$, scale the KL regularizers by $\beta = 0.1$, and bound rewards using tanh. We predict the discount factor from the latent state with a binary classifier that is trained towards the soft labels of $0$ and $\gamma$.

# B  DERIVATIONS

We define the information bottleneck objective (Tishby et al., 2000) for latent dynamics models,

$$\max \mathrm{I}(s_{1:T}; (o_{1:T}, r_{1:T}) \mid a_{1:T}) - \beta \, \mathrm{I}(s_{1:T}, i_{1:T} \mid a_{1:T}), \tag{13}$$

where $\beta$ is scalar and $i_t$ are dataset indices that determine the observations $p(o_t \mid i_t) \doteq \delta(o_t - \bar{o}_t)$ as in Alemi et al. (2016).

Maximizing the objective leads to model states that can predict the sequence of observations and rewards while limiting the amount of information extracted at each time step. This encourages the model to reconstruct each image by relying on information extracted at preceeding time steps to the extent possible, and only accessing additional information from the current image when necessary. As a result, the information regularizer encourages the model to learn long-term dependencies.

For the generative objective, we lower bound the first term using the non-negativity of the KL divergence and drop the marginal data probability as it does not depend on the representation model,

$$
\begin{aligned}
&\mathrm{I}(s_{1:T}; (o_{1:T}, r_{1:T}) \mid a_{1:T}) \\
&= \mathrm{E}_{p(o_{1:T}, r_{1:T}, s_{1:T}, a_{1:T})} \Big( \sum_t \ln p(o_{1:T}, r_{1:T} \mid s_{1:T}, a_{1:T}) - \underbrace{\ln p(o_{1:T}, r_{1:T} \mid a_{1:T})}_{\text{const}} \Big) \\
&\overset{\pm}{=} \mathrm{E}\Big( \sum_t \ln p(o_{1:T}, r_{1:T} \mid s_{1:T}, a_{1:T}) \Big) \\
&\geq \mathrm{E}\Big( \sum_t \ln p(o_{1:T}, r_{1:T} \mid s_{1:T}, a_{1:T}) \Big) - \mathrm{KL}\Big( p(o_{1:T}, r_{1:T} \mid s_{1:T}, a_{1:T}) \,\Big\|\, \prod_t q(o_t \mid s_t) q(r_t \mid s_t) \Big) \\
&= \mathrm{E}\Big( \sum_t \ln q(o_t \mid s_t) + \ln q(r_t \mid s_t) \Big).
\end{aligned}
\tag{14}
$$

For the contrastive objective, we subtract the constant marginal probability of the data under the variational encoder, apply Bayes rule, and use the InfoNCE mini-batch bound (Poole et al., 2019),

$$
\begin{aligned}
&\mathrm{E}\big( \ln q(o_t \mid s_t) + \ln q(r_t \mid s_t) \big) \\
&\overset{\pm}{=} \mathrm{E}\big( \ln q(o_t \mid s_t) - \ln q(o_t) + \ln q(r_t \mid s_t) \big) \\
&= \mathrm{E}\big( \ln q(s_t \mid o_t) - \ln q(s_t) + \ln q(r_t \mid s_t) \big) \\
&\geq \mathrm{E}\Big( \ln q(s_t \mid o_t) - \ln \sum_{o'} q(s_t \mid o') + \ln q(r_t \mid s_t) \Big).
\end{aligned}
\tag{15}
$$

For the second term, we use the non-negativity of the KL divergence to obtain an upper bound,

$$
\begin{aligned}
&\mathrm{I}(s_{1:T}; i_{1:T} \mid a_{1:T}) \\
&= \mathrm{E}_{p(o_{1:T}, r_{1:T}, s_{1:T}, a_{1:T}, i_{1:T})} \Big( \sum_t \ln p(s_t \mid s_{t-1}, a_{t-1}, i_t) - \ln p(s_t \mid s_{t-1}, a_{t-1}) \Big) \\
&= \mathrm{E}\Big( \sum_t \ln p(s_t \mid s_{t-1}, a_{t-1}, o_t) - \ln p(s_t \mid s_{t-1}, a_{t-1}) \Big) \\
&\leq \mathrm{E}\Big( \sum_t \ln p(s_t \mid s_{t-1}, a_{t-1}, o_t) - \ln q(s_t \mid s_{t-1}, a_{t-1}) \Big) \\
&= \mathrm{E}\Big( \sum_t \mathrm{KL}\big( p(s_t \mid s_{t-1}, a_{t-1}, o_t) \,\big\|\, q(s_t \mid s_{t-1}, a_{t-1}) \big) \Big).
\end{aligned}
\tag{16}
$$

This lower bounds the objective.

# C DISCRETE CONTROL

We evaluate Dreamer on a subset of tasks with discrete actions from the Atari suite (Bellemare et al., 2013) and DeepMind Lab (Beattie et al., 2016). While agents that purely learn through world models are not yet competitive in these domains (Kaiser et al., 2019), the tasks offer a diverse test bed with visual complexity, sparse rewards, and early termination. Agents observe $64 \times 64 \times 3$ images and select one of between 3 and 18 actions. For Atari, we follow the evaluation protocol of Machado et al. (2018) with sticky actions. Refer to Figure 9 for these experiments.

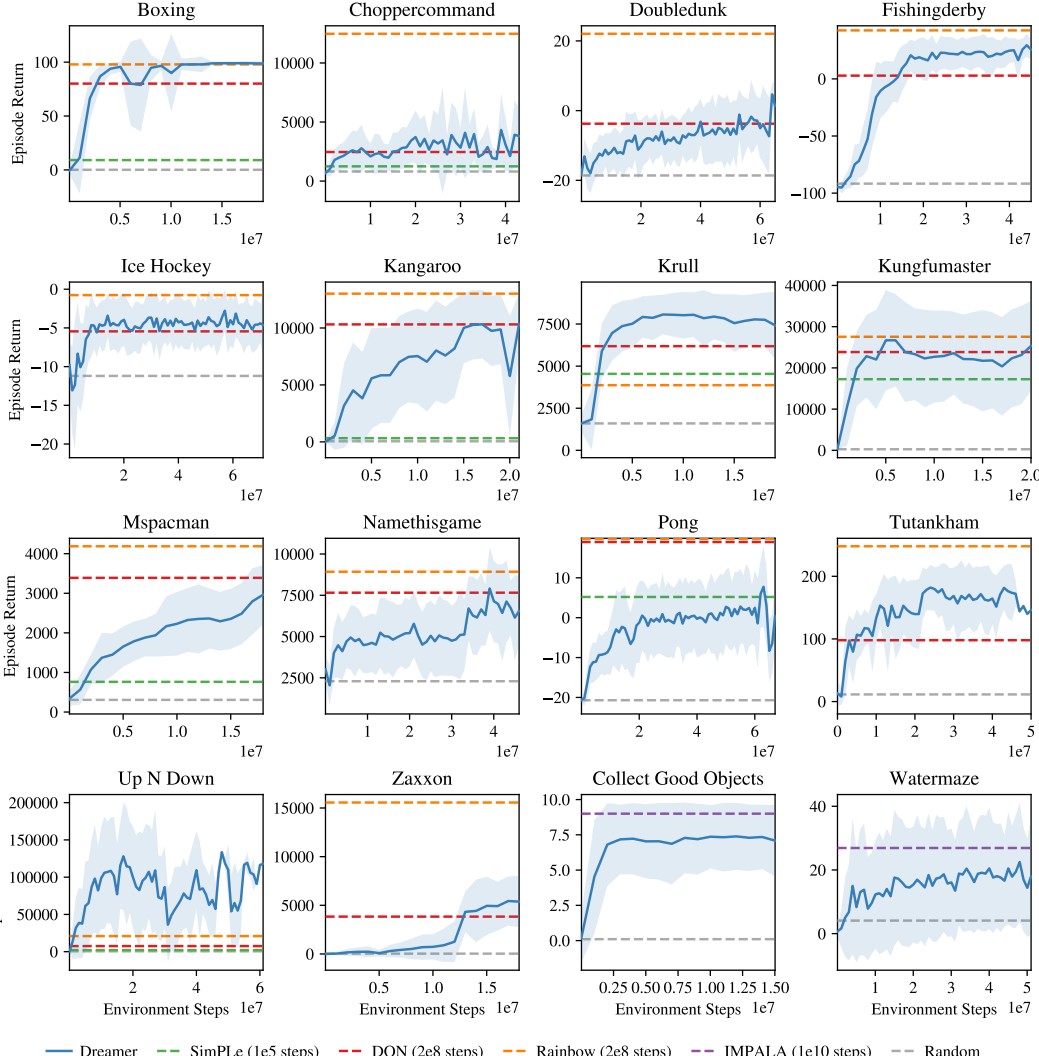

Figure 9: Performance of Dreamer in environments with discrete actions and early termination. Dreamer learns successful behaviors on this subset of Atari games and the object collection level of DMLab. We highlight representation learning for these environments as a direction of future work that could enable competitive performance across all Atari games and DMLab levels using Dreamer.

# D  BEHAVIOR LEARNING

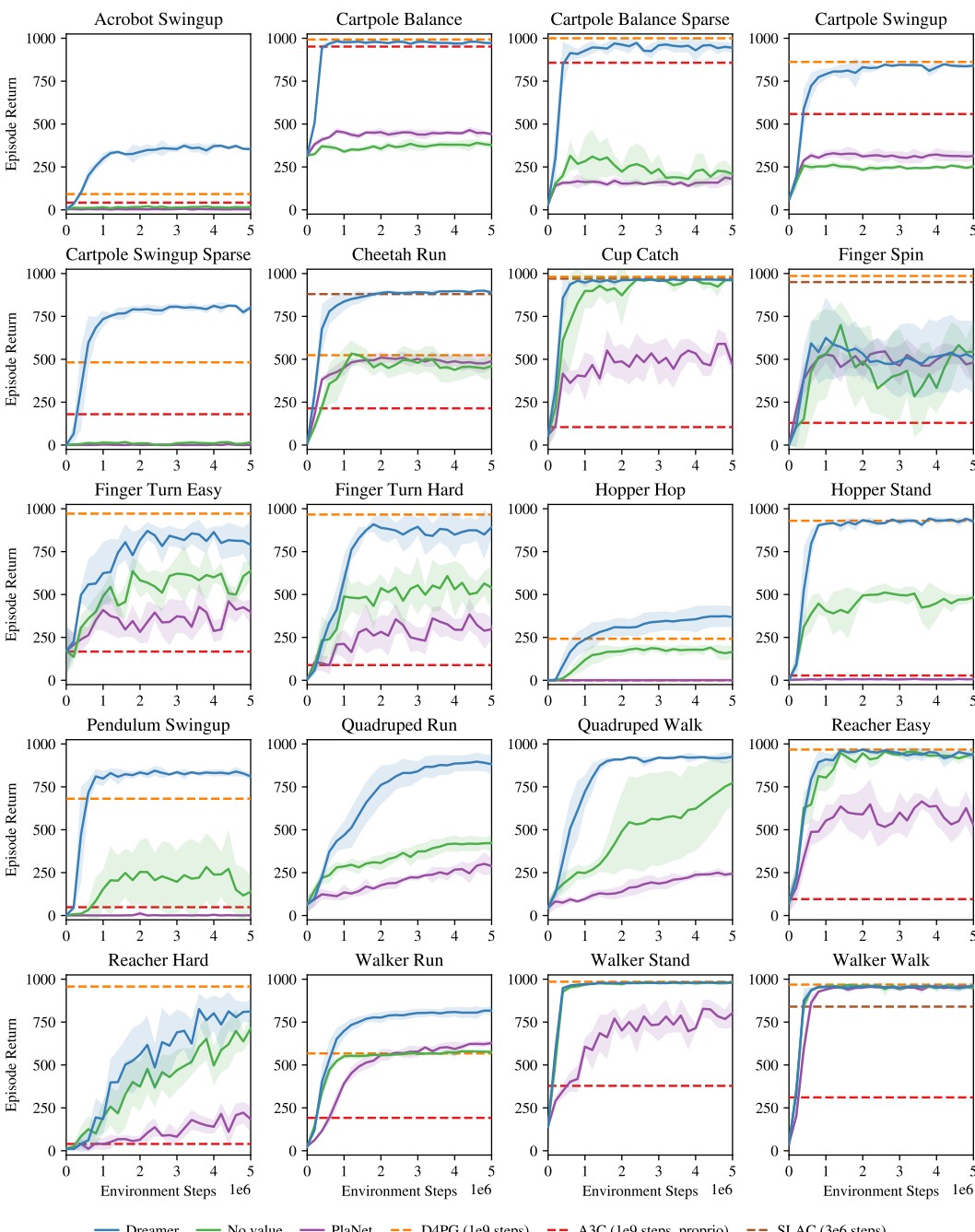

Figure 10: Comparison of action selection schemes on the continuous control tasks of the DeepMind Control Suite from pixel inputs. The lines show mean scores over environment steps and the shaded areas show the standard deviation across 5 seeds. We compare Dreamer that learns both actions and values in imagination, to only learning actions in imagination, and Planet that selects actions by online planning instead of learning a policy. The baselines include the top model-free algorithm D4PG, the well-known A3C agent, and the hybrid SLAC agent.

# E  REPRESENTATION LEARNING

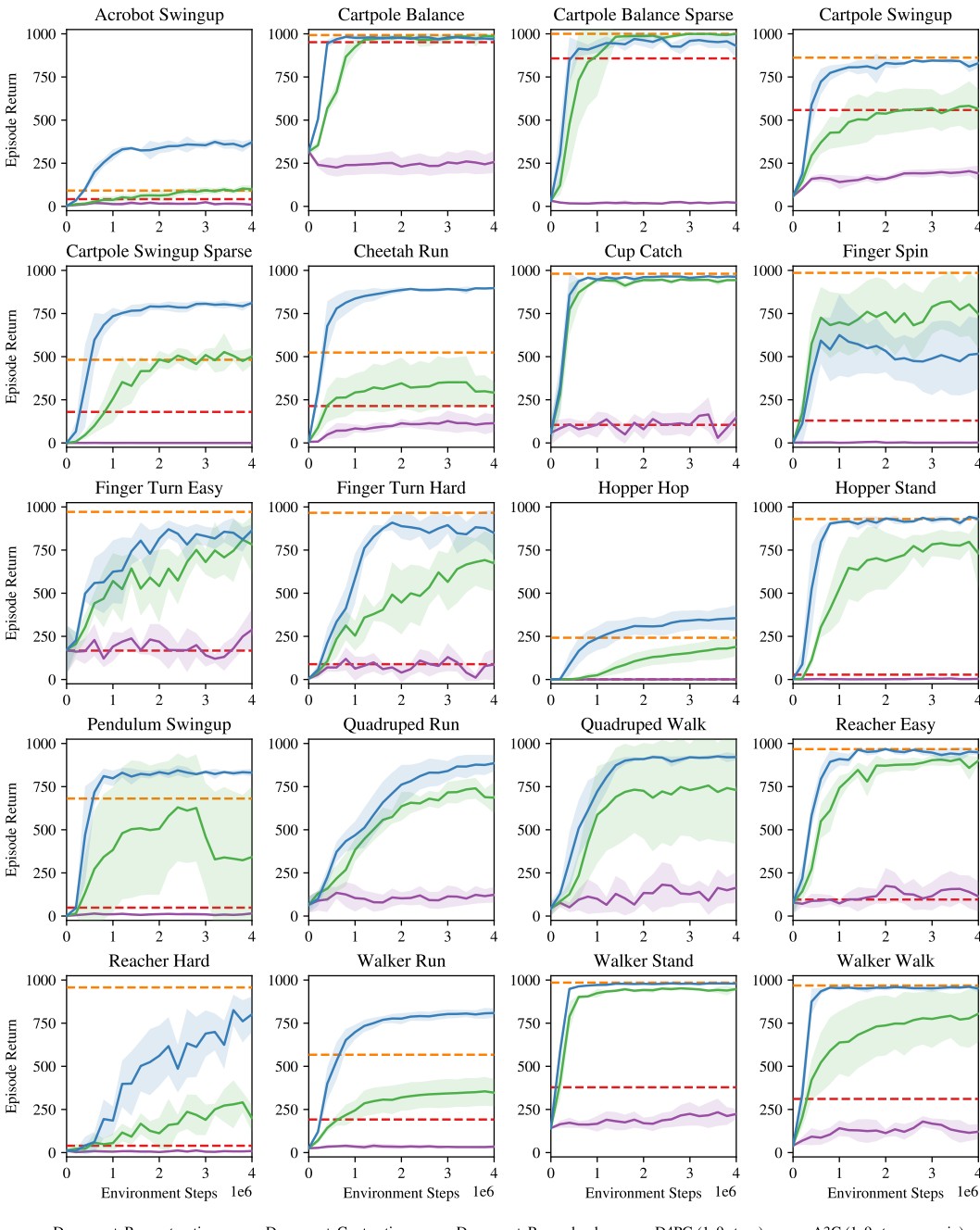

Figure 11: Comparison of representation learning methods for Dreamer. The lines show mean scores and the shaded areas show the standard deviation across 5 seeds. We compare generating both images and rewards, generating rewards and using a contrastive loss to learn about the images, and only predicting rewards. Image reconstruction provides the best learning signal across most of the tasks, followed by the contrastive objective. Learning purely from rewards was not sufficient in our experiments and might require larger amounts of experience.

# F ACTION REPEAT

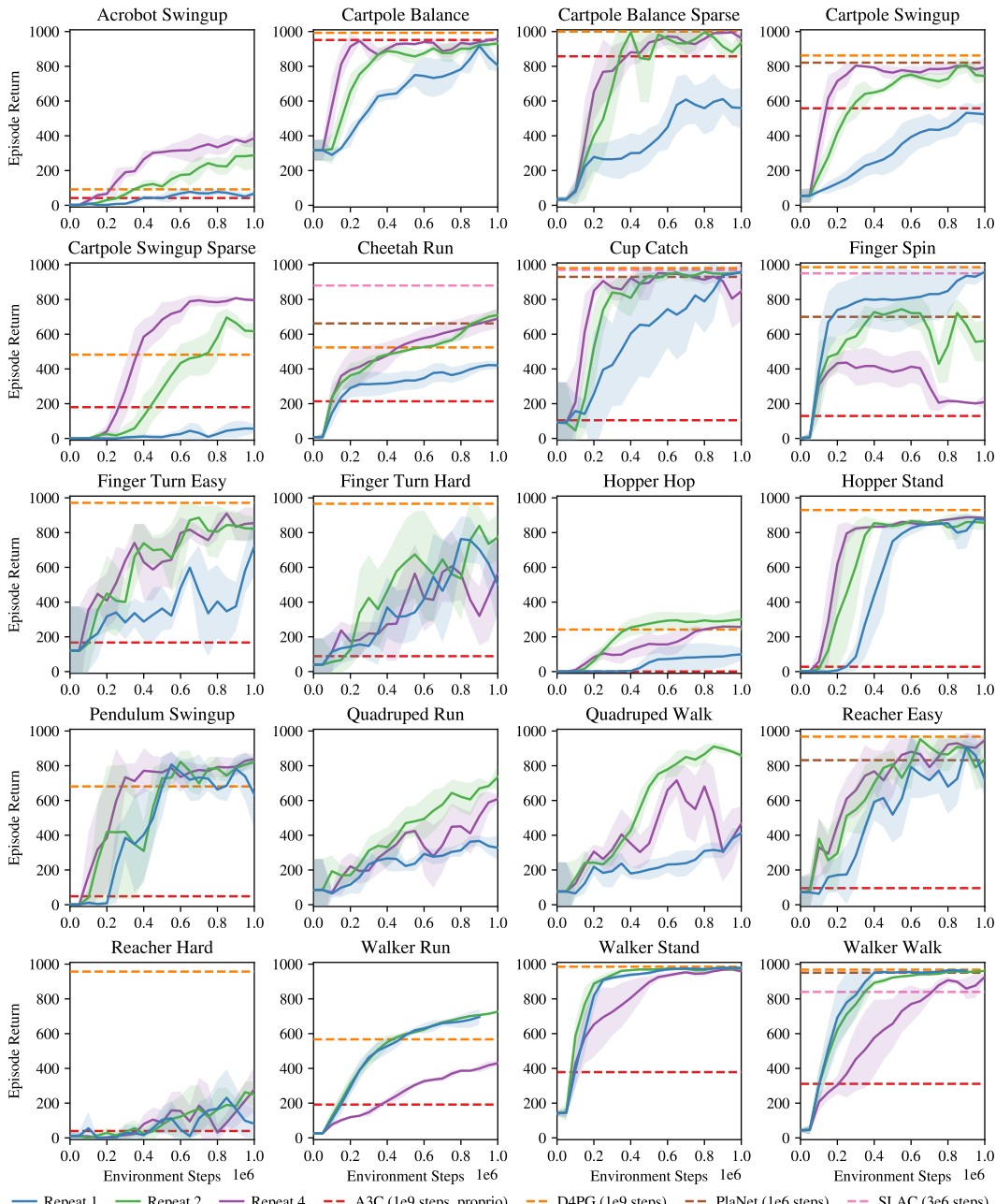

Figure 12: Robustness of Dreamer to different control frequencies. Reinforcement learning methods can be sensitive to this hyper parameter, which could be amplified when learning dynamics models at the control frequency of the environment. For this experiment, we train Dreamer with different amounts of action repeat. The areas show one standard deviation across 2 seeds. We used a previous hyper parameter setting for this experiment. We find that a value of $R = 2$ works best across tasks.

## G    Continuous Control Scores

| | A3C | D4PG | PlaNet[1] | Dreamer |
|---|---|---|---|---|
| Modality | proprio | pixels | pixels | pixels |
| Steps | $10^9$ | $10^9$ | $5 \times 10^6$ | $5 \times 10^6$ |
| Acrobot Swingup | 41.90 | 91.70 | 3.21 | **365.26** |
| Cartpole Balance | 951.60 | **992.80** | 452.56 | **979.56** |
| Cartpole Balance Sparse | 857.40 | **1000.00** | 164.74 | **941.84** |
| Cartpole Swingup | 558.40 | **862.00** | 312.56 | **833.66** |
| Cartpole Swingup Sparse | 179.80 | 482.00 | 0.64 | **812.22** |
| Cheetah Run | 213.90 | 523.80 | 496.12 | **894.56** |
| Cup Catch | 104.70 | **980.50** | 455.98 | **962.48** |
| Finger Spin | 129.40 | **985.70** | 495.25 | 498.88 |
| Finger Turn Easy | 167.30 | **971.40** | 451.22 | 825.86 |
| Finger Turn Hard | 88.70 | **966.00** | 312.55 | 891.38 |
| Hopper Hop | 0.50 | 242.00 | 0.37 | **368.97** |
| Hopper Stand | 27.90 | **929.90** | 5.96 | **923.72** |
| Pendulum Swingup | 48.60 | 680.90 | 3.27 | **833.00** |
| Quadruped Run | – | – | 280.45 | **888.39** |
| Quadruped Walk | – | – | 238.90 | **931.61** |
| Reacher Easy | 95.60 | **967.40** | 468.50 | **935.08** |
| Reacher Hard | 39.70 | **957.10** | 187.02 | 817.05 |
| Walker Run | 191.80 | 567.20 | 626.25 | **824.67** |
| Walker Stand | 378.40 | **985.20** | 759.19 | **977.99** |
| Walker Walk | 311.00 | **968.30** | 944.70 | 961.67 |
| Average | 243.70 | 786.32 | 332.97 | **823.39** |

---

[1]We re-run PlaNet with fixed action repeat of $R = 2$ to not tune the this value for each of the 20 tasks. As a result, the scores differ from Hafner et al. (2018).

