# OpenReview forum: "Dream to Control: Learning Behaviors by Latent Imagination"
_ICLR.cc/2020/Conference — Accept (Spotlight)_

### Official Review · AnonReviewer2 · 2019-10-19
**Official Blind Review #2**

**Rating:** 8

**Review:**

This work is clearly the work of a large team. the paper clearly defines what is being done. I have spent a lot of effort with MCTS. I can not find the corresponding allowance for stochastic jumps in the latent space long horizon learning.

You have the phrase "allowing to imagine thousands of trajectories in parallel". I would like some elaboration on this. I think you have ideas of what is happening in the latent space that I am not following.

You are heavy on the machinery and math. I find the learning in the latent space the important part and there are things like how much simulation is done in the latent learning not clearly spelled out. How does the effort compare to the 1E9 steps of the base line your refer to?

Your team is highly competent your style is distinct. Now may be the time to move you to understanding what structures get learned in latent space, are the in fact compact, diverse?

Perhaps there is room for memory/memories in the latent space?

Massive effort, nice results. Now for learning on our part (the humans).

**Experience Assessment:**

I have read many papers in this area.

**Review Assessment: Checking Correctness Of Derivations And Theory:**

I assessed the sensibility of the derivations and theory.

**Review Assessment: Checking Correctness Of Experiments:**

I assessed the sensibility of the experiments.

**Review Assessment: Thoroughness In Paper Reading:**

I read the paper thoroughly.

---

> ### Author Response · Authors · 2019-11-15
> **Response to Review #2**
>
> Thank you for your review!
>
> > You have the phrase "allowing to imagine thousands of trajectories in parallel". I would like some elaboration on this. I think you have ideas of what is happening in the latent space that I am not following.
>
> The latent states are defined as 330 dimensional activation vectors with 300 deterministic and 30 sampled components. We can predict imagined trajectories for thousands of initial states in parallel since they fit into the memory of the GPU at once. Specifically, Dreamer predicts imagined trajectory of length 20 from each of the 50x50=2500 latent states for the current training batch.
>
> Performing the same amount of imagination steps with a dynamics model that generates images during inference would be challenging. For example, we could only fit up to 500 trajectories of length 10 into GPU memory with the SV2P model (Babaeizadeh et al. 2017). Besides the memory constraints for predicting multiple trajectories in parallel, predictions in the latent space are often an order of magnitude faster than in pixel space.
>
> > I find the learning in the latent space the important part and there are things like how much simulation is done in the latent learning not clearly spelled out. How does the effort compare to the 1E9 steps of the base line your refer to?
>
> We will include more details in the final version. Dreamer was run for 2e6 environment steps (20 hours) compared to D4PG that was run for 1e9 environment steps (24 hours). Both algorithms used a single GPU each. As outlined in our previous answer, Dreamer performs 10 billion imagination steps throughout training. Please note that imagination steps are often considered free for robotic learning, because the bottleneck is the time of physical interaction with the real world.
>
> > [...] understanding what structures get learned in latent space, are the in fact compact, diverse?
>
> The amount of information in the latent representation is upper bounded by the KL divergence loss. We observed a typical KL divergence of 15 bits per time step, compared to the 64 x 64 x 3 x 8 bits of the corresponding images. This bounds the compression ratio to at least 1 : 6500.
>
> We make no claims regarding diversity. However, since the behaviors are learned purely in latent space, there must be a sufficient amount of diversity to solve the presented tasks. We agree that exploring the semantics of the latent space is an interesting orthogonal direction for future work.
>
> > Perhaps there is room for memory/memories in the latent space?
>
> It would be interesting to combine Dreamer with external memory modules. Gregor et al. (2019) provide a comparison of such modules. However, this would better be addressed in a separate work to keep the paper focused on the main contribution of learning behaviors by latent imagination.

---

### Official Review · AnonReviewer4 · 2019-10-21
**Official Blind Review #4**

**Rating:** 6

**Review:**

Paper summary.
The paper proposes Dreamer, a model-based RL method for high-dimensional inputs such as images. The main novelty in Dreamer is to learn a policy function from latent representation-and-transition models in an end-to-end manner. Specifically, Dreamer is an actor-critic method that learns an optimal policy by backpropagating re-parameterized gradients through a value function, a latent transition model, and a latent representation model. This is unlike existing methods which use model-free or planning methods on simulated trajectories to learn the optimal policy. Meanwhile, Dreamer learns the remaining components, namely a value function, a latent transition model, and a latent representation model, based on existing methods (the world models and PlaNet). Experiments on a large set of continuous control tasks show that Dreamer outperforms existing model-based and model-free methods.

Comments.
Efficiently learning a policy from visual inputs is an important research direction in RL. This paper takes a step in this direction by improving existing model-based methods (the world models and PlaNet) using the actor-critic approach. I am leaning towards weak accepting the paper.

I am reluctant to give a higher score due to its incremental contribution. Specifically, the policy update in Dreamer resembles that of SVG (Heess et al., 2015), which also backpropagates re-parameterized gradients through a value function and a transition model. The main difference between Dreamer and SVG is that Dreamer incorporates a latent representation model. From this viewpoint, the actor-critic component in Dreamer is an incremental contribution. Since the latent models are learned based on existing techniques, the paper presents an incremental contribution.

Besides the above comments, I have these additional comments.
- Effectiveness on very long horizon trajectories:
Simulating long-horizon trajectories with a probabilistic model is known to be unsuitable for model-based RL due to accumulated errors. This is an open issue in model-based RL. The paper attempts to solve this issue by backpropagating policy gradients through the transition model, which is known to be more robust against model errors (see e.g., PILCO (Deisenroth et al., 2011)). However, the issue still exists in Dreamer, since there seems to be an upper limit of effective horizon length (perhaps around 40, according to Figure 4). This horizon length is still short compared to the entire horizon length of many MDPs (e.g., 1000). I think this point should be discussed in the paper. That is, the issue still exists, and Dreamer is less effective with very long horizon.

- Inapplicability to discrete controls:
One restriction of re-parameterized gradients is that the technique is not applicable to discrete random variables. This restriction exists in Dreamer, and the method cannot be applied to discrete control tasks unless approximation techniques such as Gumbel-softmax are used. Still, such approximations would make learning more challenging, especially with long-horizon backpropagation. This restriction should be noted in the paper.

- There is no mention about variance of policy gradient estimates. Dreamer does not use any variance reduction technique, so the gradient estimates could have very large variance.

- q_theta was introduced in Eq. (8) before it is defined in Eq. (11). Also, I suggest moving Section 4 to be right after Section 2, since Section 4 presents existing techniques similarly to Section 2, while Section 3 presents the main contribution.


Update after authors' response.
I read the response. The paper is more clear after authors' clarification. Though, I still think the contribution is incremental, since back-propagating gradients through values and dynamics has been studied in prior works (albeit with less empirical successes compared to Dreamer). Nonetheless, I am keen to acceptance. I would increase the rating from 6 to 7, but I will keep the rating of 6 since the rating of 7 is not possible.

**Experience Assessment:**

I have published one or two papers in this area.

**Review Assessment: Checking Correctness Of Derivations And Theory:**

N/A

**Review Assessment: Checking Correctness Of Experiments:**

I assessed the sensibility of the experiments.

**Review Assessment: Thoroughness In Paper Reading:**

I made a quick assessment of this paper.

---

> ### Author Response · Authors · 2019-11-15
> **Response to Review #4**
>
> Thank you for the review and accurate summary of our submission!
>
> > I am reluctant to give a higher score due to its incremental contribution. Specifically, the policy update in Dreamer resembles that of SVG (Heess et al., 2015), which also backpropagates re-parameterized gradients through a value function and a transition model.
>
> SVG clearly differs from Dreamer in that it only considers 1-step model predictions in SVG(1) or multi-step predictions without value function in SVG(∞). SVG(0) does not use a dynamics model. In addition, Dreamer propagates gradients through transitions in a learned features, making it effective for high-dimensional control tasks.
>
> > Since the latent models are learned based on existing techniques, the paper presents an incremental contribution.
>
> Besides the important technical difference described above, we highlight the empirical performance of Dreamer. A conclusion of the SVG paper was that the model did not yield substantial practical benefits beyond 1-step predictions. We found it important to revisit this topic in the light of recent substantial improvements to dynamics models (see below).
>
> > Effectiveness on very long horizon trajectories: Simulating long-horizon trajectories with a probabilistic model is known to be unsuitable for model-based RL due to accumulated errors. This is an open issue in model-based RL.
>
> While current dynamics models still cannot accurately predict full episodes, this is rarely needed in practice. Recent works successfully use learned dynamics for control from both proprioceptive inputs (Chua et al. 2018, Shyam et al. 2019, Wang & Ba 2019) and from images (Hafner et al. 2019, Zhang et al. 2019).
>
> Dreamer shows that the relatively short model predictions (H=20) yield high-quality policy gradients, and that an additional value function in the latent space is effective for solving tasks that require longer-term credit assignment (e.g. with sparse rewards). Our experiments provide evidence that combination is effective in practice.
>
> > However, the issue still exists in Dreamer, since there seems to be an upper limit of effective horizon length (perhaps around 40, according to Figure 4). This horizon length is still short compared to the entire horizon length of many MDPs (e.g., 1000). I think this point should be discussed in the paper. That is, the issue still exists, and Dreamer is less effective with very long horizon.
>
> We address the challenge of long horizons not using long-term model predictions but by learning a value function that estimates the infinite sum of discounted future rewards. Figure 4 in our submission shows that this gives Dreamer robustness to the imagination horizon compared to two baselines.
>
> > Inapplicability to discrete controls:  One restriction of re-parameterized gradients is that the technique is not applicable to discrete random variables. This restriction exists in Dreamer, and the method cannot be applied to discrete control tasks unless approximation techniques such as Gumbel-softmax are used. Still, such approximations would make learning more challenging, especially with long-horizon backpropagation. This restriction should be noted in the paper.
>
> We applied Dreamer to environments with discrete actions using the DiCE estimator (Foerster et al. 2018) locally for the da/dμ and da/dσ derivatives. This was a drop-in replacement for the reparameterization estimator and slightly outperformed a Gumble-softmax actor. We find that with this 1 line change, Dreamer solves discrete action tasks of the Atari suite and a 3D DMLab environment.
>
> > There is no mention about variance of policy gradient estimates. Dreamer does not use any variance reduction technique, so the gradient estimates could have very large variance.
>
> Dreamer uses reparamterization gradients that already have low variance (Kingma & Welling 2013, Rezende et al. 2014); although see Miller et al. (2017). Learning baselines for variance reduction is common for Reinforce estimators as used in A3C and PPO (Mnih et al. 2016, Schulman et al. 2017) but not for reparameterization estimators as used in Dreamer, SVG, and SAC (Heess et al. 2015, Haarnoja et al. 2018).

---

### Official Review · AnonReviewer1 · 2019-10-22
**Official Blind Review #1**

**Rating:** 6

**Review:**

This paper introduced a latent space model for reinforcement learning in vision-based control tasks. It first learns a latent dynamics model, in which the transition model and the reward model can be learned on the latent state representations. Using the learned latent state representations, it used an actor-critic model to learn a reactive policy to optimize the agent's behaviors in long-horizon continuous control tasks. The method is applied to vision-based continuous control in 20 tasks in the Deepmind control suite.

Pros:
1. The method used a latent dynamics model, which avoids reconstruction of the future images during inference.
2. The learned actor-critic model replaced online planning, where actions can be evaluated in a more efficient manner.
3. The model achieved better performances in challenging control tasks compared to previous latent space planning methods, such as PlaNet.

Cons:
1. The work has limited novelty: the learning of the world model (recurrent state-space model) closely follows the prior work of PlaNet. In contrast to PlaNet, the difference is that this work learns an actor-critic model in place of online planning with the cross entropy method. However, I found the contribution of the actor-critic model is insufficient and requires additional experimental validation (see below).

2. Since the actor-critic model is the novel component in this model (propagating gradients through the learned dynamics), I would like to see additional analysis and baseline comparisons of this method to previous actor-critic policy learning methods, such as DDPG and SAC training on the (fixed) latent state representations, and recent work of MVE or STEVE that use the learned dynamics to accelerate policy learning with multi-step updates.

3. The world model is fixed while learning the action and value models, meaning that reinforcement learning of the actor-critic model cannot be used to improve the latent state model. It'd be interesting to see how optimization of the actions would lead to better state representations by propagating gradients from the actor-critic model to the world model.

Typos:
Reward prediction along --> Reward prediction alone
this limitation in latenby?

**Experience Assessment:**

I have published one or two papers in this area.

**Review Assessment: Checking Correctness Of Derivations And Theory:**

I assessed the sensibility of the derivations and theory.

**Review Assessment: Checking Correctness Of Experiments:**

I carefully checked the experiments.

**Review Assessment: Thoroughness In Paper Reading:**

I read the paper thoroughly.

---

> ### Author Response · Authors · 2019-11-15
> **Response to Review #1**
>
> Thank you for your review!
>
> > Pros:
> > 1. The method used a latent dynamics model, which avoids reconstruction of the future images during inference.
> > 2. The learned actor-critic model replaced online planning, where actions can be evaluated in a more efficient manner.
> > 3. The model achieved better performances in challenging control tasks compared to previous latent space planning methods, such as PlaNet.
>
> This is an accurate summary. We would like to highlight two additional points. First, the improved performance is attributed to a novel actor-critic algorithm that uses analytic multi-step gradients of predicted state-values (not Q-values). Second, in addition to outperforming previous latent space planning methods, the proposed algorithm also outperforms the model-free D4PG algorithm, the previous state-of-the-art on this benchmark suite.
>
> > 1. The work has limited novelty: the learning of the world model (recurrent state-space model) closely follows the prior work of PlaNet. In contrast to PlaNet, the difference is that this work learns an actor-critic model in place of online planning with the cross entropy method. However, I found the contribution of the actor-critic model is insufficient and requires additional experimental validation (see below).
>
> Dreamer is a novel algorithm that belongs to the family of actor critic methods. At a high level, previous approaches can be grouped into those using Reinforce gradients with V baselines (A3C, PPO, ACER) and those using deterministic or reparameterization gradients of learned Q functions (DDPG, SAC, MVE, STEVE). In comparison, Dreamer uses reparameterization gradients of V functions by backpropagating the value estimates through the latent dynamics.
>
> Specifically, while Reinforce estimators typically learn V functions, these are only used to reduce the variance of the gradient estimate rather than directly maximizing them with respect to the actor. Actor-critic algorithms that use analytic gradients of Q critics differ from Dreamer in two ways. First, they learn a Q function rather than just a V function. Second, the actor only maximizes the Q value predicted for the current time step rather than maximizing multi-step value estimates.
>
> While MVE and STEVE learn dynamics models (from proprioceptive inputs), the dynamics are not directly used to update the policy. Instead, they only serve for computing multi-step Q targets for learning the Q critic. Thus, no gradients are backpropagated through the dynamics model for learning the actor or critic. Please also see the comparison in the last paragraph of Section 3, which we will extend with the present discussion.
>
> > 2. Since the actor-critic model is the novel component in this model (propagating gradients through the learned dynamics), I would like to see additional analysis and baseline comparisons of this method to previous actor-critic policy learning methods, such as DDPG and SAC training on the (fixed) latent state representations, and recent work of MVE or STEVE that use the learned dynamics to accelerate policy learning with multi-step updates.
>
> As summarized above, Dreamer differs from previous actor-critic algorithms not just by using latent dynamics but also by using analytic multi-step gradients of a V function rather than one-step gradients Q function. This renders Dreamer conceptually distinct from DDPG, SAC, MVE, and STEVE.
>
> We have run experiments with MVE in the latent space of the same dynamics model and tuned the learning rate for actor and Q function. We did not find an improvement over Dreamer (MVE worked worse across tasks) in these experiments, possibly because it only updates the Q function at the initial state of the imagination rollout.
>
> Note that with a model, Q values can be computed by combining the dynamics with a value function, so learning Q is not necessary anymore. Since using V in Dreamer outperforms the state-of-the-art D4PG agent and is simpler than the Q function in DDPG and MVE and substantially simpler than STEVE (ensemble of models) and SAC (two Q functions, one V function), we argue for this design choice.
>
> > 3. [...] It'd be interesting to see how optimization of the actions would lead to better state representations by propagating gradients from the actor-critic model to the world model.
>
> We have run these experiments and it prevented learning completely. Using gradients of the action or value models to shape the dynamics allows them to "cheat". Specifically, the actions maximize value estimates; using these to update the dynamics results in overly optimistic dynamics. The values maximize Bellman consistency; using these to update the dynamics can encourage collapse of the latent space. As a result, we suggest the perspective of viewing the dynamics as a fixed MDP during imagination training. We will add a discussion of this to the paper.
>
> If we addressed your concerns satisfactorily, we would be happy if you would consider updating your score.

---

### Official Review · AnonReviewer3 · 2019-10-25
**Official Blind Review #3**

**Rating:** 8

**Review:**

This paper presents a world model-based approach in which behaviours are optimised by rollouts (i.e. imagination) in latent space. The paper achieves impressive results across a large selection of tasks, both in terms of sample efficiency and final performance.

I found the paper interesting to read and well written. The main contribution (backpropagating analytic gradients through imagined trajectories?) could potentially be highlighted more but otherwise the paper was clear. I wonder if the authors ever looked at how much the size of the latent vector determines the performance of the system? Is there an optional latent vector size across domains or is that optimal size task dependent?

Additionally, how much variance is there in the imagined trajectories from a certain starting state? In other words, are the endpoints of most imagined trajectories similar or very different?

There is actually not too much for me to critique and I would suggest this paper should be accepted.


Minor comment:
- On page 2 it says “We approach this limitation in latenby”, which I assume is a typo?

####After rebuttal####
The authors' response addressed my remaining questions.


**Experience Assessment:**

I have published one or two papers in this area.

**Review Assessment: Checking Correctness Of Derivations And Theory:**

I did not assess the derivations or theory.

**Review Assessment: Checking Correctness Of Experiments:**

I assessed the sensibility of the experiments.

**Review Assessment: Thoroughness In Paper Reading:**

I read the paper at least twice and used my best judgement in assessing the paper.

---

> ### Author Response · Authors · 2019-11-15
> **Response to Review #3**
>
> > I found the paper interesting to read and well written. The main contribution (backpropagating analytic gradients through imagined trajectories?) could potentially be highlighted more but otherwise the paper was clear.
>
> Thank you. Correct, our main contribution is to learn long-horizon behaviors by propagating analytic value gradients through imagined trajectories. Moreover, we show that this yields a scalable algorithm that solves control tasks of higher difficulty than was previously possible using model-based agents.
>
> > I wonder if the authors ever looked at how much the size of the latent vector determines the performance of the system? Is there an optional latent vector size across domains or is that optimal size task dependent?
>
> For our experiments, we used the same hyper parameters across all tasks, including the state size. We conducted an additional experiment where we trained Dreamer with latent states of 100, 200, 300, 400, 500 deterministic units and 10, 20, 30, 50, 100 stochastic units. We find that all sizes equal to or larger than the 200 and 30 used in our main experiments yield very similar performance, while smaller sizes result in suboptimal scores on some of the tasks, hinting at insufficient model capacity.
>
> > Additionally, how much variance is there in the imagined trajectories from a certain starting state? In other words, are the endpoints of most imagined trajectories similar or very different?
>
> We have not studied this quantitatively. Qualitatively, we find more diversity in the stochastic multi-step predictions near states that are more challenging to predict. For example, this includes collisions with the ground, the unstable equilibrium of an upright balanced pendulum that could either rotate left or right, or cheetah balancing on its front feet which might flip over on its back or fall back on its feet.
>
> > There is actually not too much for me to critique and I would suggest this paper should be accepted.
>
> Thank you.

---

### Comment · AnonReviewer2 · 2019-10-18
**data efficiency**

In performance comparison you mention 2 million environment steps. I can not tell how many latent imagined steps if any? Is it zero? Or many?

You talk about representation learning, I think there are several ways to think about representation. There is the way Paul Cisek of The University of Montreal talks about feedback loops that maintain homeostasis via interaction with the environment. These have no explicit representation they are simple their own meaning based on their outcome. Then there is abstract thinking with internal visualization, planning, goal setting. humm maybe abstract representations are just more [across several variables/modes] of the same.

Can your agent learn compact representations of long horizon latent trajectories? Does it just memorize long paths?

Is there a memory? I am guessing no. Should there be a memory? How much memory is needed I wonder. Your learned  long paths do they just exist as vague imprints in the weights? Can they be made more explicit?

It is nice to read a paper where the author have a good command of English.

---

> ### Author Response · Authors · 2019-10-21
> **Thank you for the questions**
>
> Dreamer imagines ahead only during training between episodes. The 2 million environment steps correspond to 2000 episodes. Before each episode, the agent performs 100 gradient updates. For each update, the agent randomly draws a mini batch of 50 subsequences of length 50 from its past experience. From each corresponding latent state, we imagine a sequence of horizon length 20. This results in a total number of 10 billion imagination steps. In comparison, PlaNet [1] uses 120 billion imagination steps. We will include these details in the paper.
>
> We agree that there are interesting perspectives on the nature of representation learning in the neuroscience literature. That said, this is not the focus of our submission.
>
> We used world models without external memory modules. These world models remember information by passing it along the chain of compact latent states. A recent comparison of memory modules can be found in Gregor et al. [2]. Augmenting Dreamer with external memory is an orthogonal direction that we leave for future work.
>
> [1] Hafner et al., Learning Latent Dynamics for Planning from Pixels. ICML, 2019.
> [2] Gregor et al., Shaping Belief States with Generative Environment Models for RL. arXiv:1906.09237, 2019.

---

> > ### Comment · AnonReviewer2 · 2019-10-21
> > **Thank you**
> >
> > Thank you for your thoughtful and informative answers.

---

### Public Comment · ~Daniel_Mouritzen1 · 2020-01-29
**Mistake in eq. 6**

Thanks for another great paper. I noticed a mistake in in the $\lambda$-return, equation 6: The exponents for the $\lambda$ terms should have a $-1$, like so:
$$
\mathrm{V}_λ \left(s_τ\right) ≜ (1 - λ) \sum\limits_{n = 1}^{H - 1} λ^{n-1} \mathrm{V}_{\mathrm{N}}^n \left(s_τ\right) + λ^{H-1} \mathrm{V}_{\mathrm{N}}^H \left(s_τ\right)
$$
This ensures the weights sum to 1 (and it's also consistent with Sutton et al., eq. 7.6).

---

> ### Author Response · Authors · 2020-02-11
> **Thanks**
>
> Thanks, it'll be fixed in the camera-ready version.

---

### Public Comment · ~Qibing_Li1 · 2020-03-21
**Core ideas of equations (7)(8) are quite similar to a previous paper**

Core ideas of equations (7)(8) are quite similar to a previous paper, which optimizes the action policy and captures predictive representations by jointly minimizing the imagination error of simulated purchase trajectories, while maximizing the extrinsic purchase reward.

Title: PURCHASE AS REWARD: SESSION-BASED RECOMMENDATION BY IMAGINATION RECONSTRUCTION
Link: https://openreview.net/forum?id=SkfTIj0cKX

It is reviewed in ICLR 2019, but not cited in your paper. Every idea deserves respect even though it is not accepted.

---

### Decision · Program_Chairs · 2019-12-19

**Decision:**

Accept (Spotlight)

**Comment:**

This paper presents an approach to model-based reinforcement learning in high-dimensional tasks. The approach involves learning a latent dynamics model, and performing rollouts thereof with an actor-critic model to learn behaviours. This is extensively evaluated on 20 visual control tasks.

This paper was favourably received, but there were concerns around it being incremental (relative to PlaNet and SVG). The authors highlighted the differences in the rebuttal, clarifying the novelty of this work.

Given the interesting ideas presented, and the convincing results, this paper should be accepted.